# Care and Safety of Schoolchildren with Type 1 Diabetes Mellitus: Parental Perceptions of the School Nurse Role

**DOI:** 10.3390/healthcare10071228

**Published:** 2022-06-30

**Authors:** Marianna Drakopoulou, Panagiota Begni, Alexandra Mantoudi, Marianna Mantzorou, Georgia Gerogianni, Theodoula Adamakidou, Victoria Alikari, Ioannis Kalemikerakis, Anna Kavga, Sotirios Plakas, Georgia Fasoi, Paraskevi Apostolara

**Affiliations:** 1Department of Nursing, Faculty of Health Sciences, University of West Attica, 122 43 Athens, Greece; amantoudi@uniwa.gr (A.M.); mantzorou@uniwa.gr (M.M.); ggerogiani@uniwa.gr (G.G.); thadam@uniwa.gr (T.A.); vicalikari@uniwa.gr (V.A.); ikalemik@uniwa.gr (I.K.); akavga@uniwa.gr (A.K.); skplakas@uniwa.gr (S.P.); gfasoi@uniwa.gr (G.F.); vapostolara@uniwa.gr (P.A.); 21st Health Center of Salamina, 189 00 Salamina, Greece; yiotabegni@gmail.com

**Keywords:** type 1 diabetes mellitus, school nursing, school-based care, school-based safety, parental perceptions

## Abstract

Schoolchildren with type 1 diabetes mellitus (T1DM) need supervision in the management of their disorder by the school nurse, securing proper care and safety in the school environment. The aim of this study was to investigate the parents’ perceptions regarding the care and safety of their children with T1DM at school. In this cross-sectional study, 356 parents of children with T1DM attending primary and secondary school (convenience sample) completed the “Parents’ Opinions about School-based Care for Children with Diabetes” and the “Safety of children with T1DM at school”. The majority (58.8%) noted that their children received some care from a school nurse, less than half (44.6%) declared feeling very safe concerning diabetes care, and 42.5% reported high levels of diabetes management satisfaction. Younger age of the child (*p* < 0.001), school nurses’ advanced diabetic care skills (*p* < 0.001), existence of school nurse’s office (*p* < 0.05) and higher educational level of the father were positively correlated with higher parental feelings of safety and satisfaction. The presence of a school nurse was associated with higher academic performance (*p* < 0.001), significantly fewer absences due to the disorder (*p* < 0.001) and better diabetes management (*p* < 0.043). The daily presence of a school nurse in school decreases absenteeism, greatly improves school performance and enhances diabetic management of schoolchildren with T1DM.

## 1. Introduction

Type 1 diabetes mellitus (T1DM) is the commonest metabolic disorder in children [1]. It is a chronic autoimmune disease that leads to progressive destruction of the β pancreatic cells and culminates in non-secretion of insulin and persistent hyperglycemia [2]. Diabetes care requires lifestyle adjustments and a daily management routine [3], including the time spent in school [4], which may be up to seven hours daily [5,6]. Proper management of T1DM daily at school is associated with improved metabolic control, minimization of the risk of hypoglycemia and, in general, better quality of life, as it reduces the risk of microvascular complications or delays their development [7]. In addition, T1DM is associated with low academic achievement due to difficulties in glucose control, absenteeism and lack of concentration [8]. The school nurse’s presence guarantees the proper management of metabolic control, as well as the safety and also the academic development of students who face special health problems such as T1DM [9].

A sense of security at school is important for adjustment and progress. Students’ high self-esteem is associated with a strong sense of security at school [10,11,12]. In general, a safe school environment improves the educational experience and enhances the well-being and health of students [13]. The strategies that make school a safe environment focus on health education so that students may adopt behaviors that promote health and safety. Schools are, therefore, called upon to play an important role in promoting health and preventing illness and injury [14]. However, in a recent US study, it was found that the school staff was not well trained and did not have confidence in their diabetes management skills [15].

A 2017 qualitative study in Brazil revealed that successful management of diabetes requires the cooperation of school nurses, school staff and families of children with T1DM [16]. Although families largely determine the development of health-promoting behaviors, the school nurse can reinforce desirable behaviors and dampen down unwanted ones [17]. It thus emerges that there is a need for nursing care of the student with T1DM in the school environment, so that the school on the one hand is a safe environment and on the other offers the same level of access to educational opportunities to all children [18].

However, despite global research affirming the benefits of school nursing, there are developed countries where the majority of schools do not have school nurses [19], but even among countries that implement school nursing, there is variability in the practice of school nursing [20]. In addition to providing care to students in emergencies, the impact of providing integrated school nursing was investigated in a 2017 study in Turkey that concluded that integrated school nursing contributed positively to students’ academic performance [21]. In schools where students report fewer problems, higher academic goals are set by them [10]. A similar study in public high schools in America found a possible link between students attending schools with a full-time school nurse and significantly higher graduation rates, lower student absenteeism and higher college enrollment scores. This, according to the researchers, was due to school nursing improving health indicators and at the same time contributing to the improvement of academic achievement [22]. During the turbulent period of adolescence, the management of T1DM becomes even more difficult. A 2017 survey in the US identified concerns that focused, among other things, on the lack of full-time nurses in schools, a lack of relevant T1DM teacher information, the lack of access among diabetic children to self-care supplies, and limited communication between parents and school staff about diabetes, resulting in diabetes having an impact on long-term well-being in adulthood [7]. The presence of a school nurse ensures the safety of all schoolchildren, both healthy and those facing health problems, such as T1DM, and, in this instance, guarantees proper management of metabolic control and academic development of children with diabetes [9].

In Greece, school nurses are usually employed in special education schools and private schools. According to the legislature, school nurses can also be employed in general public primary schools at the specific request of the parents of children with special healthcare needs. A medical diagnosis and report from a public hospital doctor is a necessary requirement, in which the need for the support of a school nurse has to be documented. So, a school nurse in general public education is annually employed in primary schools only if there are children with a special healthcare needs diagnosis [23].

Limited studies have been conducted in Greece to explore the views of parents of schoolchildren with T1DM regarding the care and safety provided by a school nurse to their children in the school environment. The present study sought to highlight the problems faced by diabetic children in school—and consequently by Greek families—throughout the country, as well as the role of school nurses in the management of T1DM in the school community, and to highlight their important contribution to the wider Greek society.

## 2. Materials and Methods

### 2.1. Instruments

In the present cross-sectional correlational descriptive study, a demographic and clinical characteristics questionnaire of the participants and the following two scales were used:

“Parents’ Opinions about School-based Care for Children with Diabetes” by Driscoll et al. [24].

This is a 38-item questionnaire with sections on information about the child, the diabetes care training of the school personnel, parental opinions about the child’s safety in school, staff handling of low or high blood glucose incidents at school, the medical staff in the child’s school and school policies on managing diabetes.

“Survey for safety of children with T1DM at school” by Alaqeel [25]. This is a 30-item questionnaire examining diabetic management at school, negative peer comments, absenteeism and school performance for children with T1DM.

Participants were able to answer each question regarding their opinions on a 5-point Likert scale.

Cultural weighting of both the questionnaires for adaptation to the Greek language was performed following permission for use from the authors. A reverse translation of the questionnaire was performed, from English to Greek and vice versa by two independent translators following the standard guidelines for questionnaire translation to the Greek language.

Prior to the main study, a pilot study was conducted on nine parents of schoolchildren with T1DM in order to check for any difficulties in comprehension of the questionnaires’ content. Based on the results of this pilot study, no modifications were made. The data obtained from these questionnaires were s not included in the study.

### 2.2. Participants and Data Collection

The sample consisted of 356 parents (288 mothers and 68 fathers) with a child with type 1 diabetes. As there are no school nurses in general public schools, only parents of children with special healthcare needs are familiar with the concept of school nursing. So, an online platform was created through Google forms for participation in the study and was made available to six (6) associations and two (2) online groups of parents of children with T1DM in primary and secondary education, from all parts of the country. Inclusion criteria were (a) being over 18 years of age, (b) being parents of children with T1DM in primary (6–12 years) or secondary (13–18 years) education, and (c) having an oral and written understanding of the Greek language.

For the realization of the study, permission was requested from the boards of the associations of parents of schoolchildren with T1DM, and their consent was secured. Permission was also sought from and granted by the website and group administrators, following which the questionnaire was disseminated. A convenience sample of parents of schoolchildren with T1DM was collected from March 2020 until May 2020. The study conformed to the rules of ethics that govern the research process.

## 3. Statistical Analysis

Quantitative variables were expressed as mean values (standard deviation) and as median (interquartile range), while qualitative variables were expressed as absolute and relative frequencies. Mann–Whitney test was used for the comparison of continuous variables between two groups. For the comparison of proportions, chi-square and Fisher’s exact tests were used. Logistic regression analysis in a stepwise method (*p* for entry 0.05, *p* for removal 0.10) was used in order to find independent factors associated with parents feeling very/extremely safe with the care provided to their children during a normal school day and parents feeling very/extremely satisfied with the care provided to their children during a normal school day as dependent variables. Adjusted odds ratios (OR) with 95% confidence intervals (95% CI) were computed from the results of the logistic regression analyses. All reported *p* values are two-tailed. Statistical significance was set at *p* < 0.05, and analyses were conducted using SPSS statistical software (version 22.0).

## 4. Results

The demographic characteristics of the parents and children are presented in Table 1.

The mean age of the children was 11 years (SD = 4.0 years), and 55.7% were girls. Mean age at diagnosis was 5.9 years (SD = 3.4 years), and 93.2% of the children attended public school. The average number of students in each class was 20.7 (SD = 5.5).

Information regarding the care of the child’s diabetes is presented in Table 2.

About one out of ten (11.0%) children attended extended school, and in most cases, the children were responsible for their own diabetic management. In 34.4% of cases that did not attend extended school, it was due to diabetes. Additionally, 77.1% of children were allowed to check their glucose levels anywhere in the school, and 68.6% were allowed to inject insulin anywhere in the school. Further, 94.4% of children were allowed to use the toilet facilities whenever needed, and 94.9% had snacks that could be consumed during class. Most parents (85.9%) provided their children with medical supplies and snacks for diabetic management at school. In 58.8% of cases, there was a school nurse, and in 21.9%, there was a school nurse’s office in school. Moreover, 89.9% of the participants knew how to get in touch with the person taking care of their child’s diabetes at school, and 78.1% considered the role of the school nurse very important.

Furthermore, 44.6% of participants felt that their child was very/extremely safe during a normal school day (Table 3).

The parents’ satisfaction with the handling of a low or high glucose incident was significantly higher in cases where their children’s school had a nurse’s office. Furthermore, the number of absences due to diabetes was significantly lower in cases where their children’s school had a nurse’s office. Additionally, the child’s school performance and most recent HbA1c values were better in the instances where a school nurse was present.

Moreover, 42.4% of participants were very/extremely satisfied with the care provided for their children during a normal school day (Table 4).

The percentage of parents who felt very/extremely safe with the care provided to their child during a normal school day was significantly lower when their child or the parents themselves provided most of the care, while it was significantly higher when most of the care was provided by a school nurse. In addition, parents who felt very/extremely safe with the care provided to their child during a normal school day rated their teachers’ and school nurse’s diabetic care skills higher than those who felt not at all/a little/moderately safe. The percentage of parents who felt very/extremely safe with the care provided to their child during a normal school day was significantly lower among those who felt that the principal should be trained in the care and needs of children with diabetes and significantly higher among those who believed that some “other” school employee should be trained. In addition, the percentage of parents who felt very/extremely safe with the care provided to their child during a normal school day was significantly lower among those who preferred that the principal provide care and significantly higher among those who preferred that it be provided by the teacher or some “other” school employee. The percentage of parents who felt very/extremely safe with the care provided to their child during a normal school day was significantly lower when their child knew how to administer insulin without any supervision or assistance.

The age of the child, the parents’ perception of the school nurse’s level of ability in providing care for the child with diabetes during a normal school day, the existence of a school nurse’s office and the father’s level of education were found to be independently associated with parents feeling very/extremely safe and very/extremely satisfied with the care provided to their child (Table 5).

Specifically, the older the children were, the less likely it was that the parents would feel very/extremely safe and be very/extremely satisfied with the care provided to their child during a normal school day. In the cases where the school had a school nurse’s office, the probability that parents would feel very/extremely safe and very/extremely satisfied with the care provided to their child during a normal school day was 3.64 and 2.40 times higher, respectively. The more able they considered the school nurse to care for their child’s diabetes during a normal school day, the more likely it was that the parents would feel very/extremely safe and very/extremely satisfied. When the father was a high school or college graduate, the probability that the parents would feel very/extremely satisfied with the care provided to their child during a normal school day was 3.71 times higher compared to when his educational level was primary/middle school.

## 5. Discussion

The prevalence of T1DM is increasing annually in Europe and in Greece [26,27], creating thousands of new cases each year. Therefore, the proper management of the diabetic child in school poses a challenge for school nurses.

An interesting finding of our study concerns the levels of parental perception of their children’s safety and care during the school day. Parents felt safer when the nurse provided most of the care during the school day, compared to the proportion of parents who felt safer when the majority of care was provided by themselves or the child. This is supported by Wilt, who linked the security and satisfaction that parents feel with the school nurse/student ratio. The study showed that parents felt less secure and less satisfied when their children attended schools with higher nurse/student ratios [9].

The literature confirms that the presence of a full-time school nurse is vital for parents [7,28,29], as the school body is considered responsible for the health of each student, especially in primary school [30,31]. Furthermore, teachers also feel safer with the presence of a school nurse according to a study conducted in Greece. Teachers and other health professionals consider the school nurse an important member of the team [32]. School nurses do not only focus on children with diabetes, but on promoting student health in general [33]. The health and well-being of children at school is provided by the school nurse as a member of the healthcare team [34,35]. In the present study, in 58% of cases there was a school nurse to care for the child with diabetes. The results are similar in almost all studies worldwide. In Israel, a school nurse is only available a few days a month [18], or is hired only in a special school group, as is the case in private schools in Ireland [36], while in Brazil, the presence of school nurses is vital because teachers are not required to administer insulin [16]. Similarly in the US, depending on the state, a school nurse is hired, or else non-medical staff are allowed to provide care to the diabetic child [25]. In the state of Alabama, the school nurse is the only staff member authorized by law to administer medication. In general, parents expressed concern about the quality of care in the absence of a school nurse, while many believed that the constant presence of a school nurse would improve student care and reduce their concerns [37]. Contrarily, in another US state, if a school nurse was unavailable, they alternatively suggested the training of non-medical staff, stating that the parents feel just as safe [24]. Recent research, however, has shown that non-medical staff assigned to care for children with diabetes at school are not well trained and also do not feel confident in their ability to manage children with T1DM at school [15].

At the same time, almost half of our study participants were very satisfied with the care their children received during the school day. In two separate studies in the US, the results varied significantly; the highest rate of satisfaction (83.1%) was reported by parents in Skelley et al. [37], while in the Jacquez et al. study, parents found a lack of support from school and expressed concerns about the care their children received for their diabetes [28]. Conversely, an important finding was that in cases where children with diabetes were cared for by a school nurse, the levels of glycosylated hemoglobin were significantly lower. Extensive research has shown that optimal regulation of T1DM can prevent or delay the onset of chronic complications [7]. Researchers have also identified deficits in communication between schools, parents and health care providers and highlighted the necessity of adequate support, especially for adolescents with T1DM [38,39]. It is encouraging that 91% of the children in the study used insulin at school, and the school nurse was described by the majority of parents as “very capable” of delivering insulin to their children. The presence of a school nurse and the implied care enhances the sense of security of children in the school environment and improves their school performance as well as their metabolic regulation [40]. In contrast, in another study, only 34% of parents believed that a mild hypoglycemic episode could be recognized by teachers [38].

Other important findings were that children were allowed to control their blood glucose throughout the school (77.1%) and inject insulin anywhere in school (68.6%), and the majority were allowed to use the toilet facilities whenever needed. It has been shown that school rules usually do not facilitate the self-management of T1DM in school [41], as children are not allowed to control their blood glucose levels or inject insulin in the classroom [28]. Schwartz et al., in their study, report that almost half of children with diabetes experienced at least one incident of interference with diabetes self-care or restricted toilet use at school [29].

Our results also showed that parental satisfaction was higher when the father was a high school or college graduate, compared to when he was a primary/middle school graduate. This may mean that parents with higher academic achievements had better levels of communication with the school nurse and had knowledge of and appreciation for their role. In Wilt’s study (2020), parental feelings of safety and satisfaction were higher than in previous studies and were associated with higher parental educational levels, similarly to our study [9].

The importance of the school nurse’s role is paramount, as learning and health are interrelated. The average number of absences due to diabetes in this study was 6 days per year; however, the number of absences differs significantly between educational levels. Specifically, in high schools in Greece, where the majority do not have a school nurse, there were more absences. Half of the schoolchildren with chronic health problems were absent from school on a regular basis, and an additional 10% missed more than 25% of the school year [7]. T1DM is associated with low academic achievement due to glucose control difficulties, absenteeism, and lack of concentration [8].

A school nurse’s presence is important for monitoring diabetic self-management in older schoolchildren, which ultimately promotes safety in the school environment [42]. Our study also revealed the fact that the older the children were, the less likely the parents were to feel safe and satisfied with the care of their children at school, since in the majority of high schools in Greece, the institution of the school nurse does not exist. In addition, another study found that the older the adolescent, the less compliant they were with diet and glucose control, and the longer the diabetes was established, the worse the metabolic control was [43].

### Limitations

A limitation of cross-sectional studies is the inability to make a causal inference; thus, identified associations may be difficult to interpret. Moreover, the parents who participated in the present study were members of associations for T1DM, so they were informed about the role of the school nurse. Parents had to have email accounts or be members of social networking groups related to juvenile diabetes and be active on social media. HbA1c levels were reported by parents and not measured in one laboratory and may have varied.

It would be interesting to explore the perceptions of parents who are not members of associations and whose awareness is not as high. An example of such a population could be parents whose children attend outpatient diabetic clinics in public pediatric hospitals.

## 6. Conclusions

The sense of safety and satisfaction that parents of schoolchildren with T1DM have regarding the care their children receive in the school environment has been studied internationally in recent years, demonstrating the concern of parents about the diabetic profile of children at school. The presence of school nurses as permanent staff members in every school is considered necessary, as their daily presence helps to significantly reduce the absenteeism of diabetic schoolchildren, significantly increases their school performance and, at the same time, contributes to better regulation of T1DM.

A determining factor for better integration of the child with diabetes into the school environment is to increase the number of school nurses. According to Greek legislation, the recruitment of a school nurse is realized at the request of the parent and only following a medical diagnosis from a public hospital doctor. The school nurse’s presence is necessary in all levels of education, since adolescents have significant diabetes management needs that increase due to the psychological burden they experience. National policies need to be reinforced so that schools in Greece will permanently employ school nurses in order to ensure that school is a safe environment for schoolchildren with diabetes, promoting learning and health. In the words of Lina Rogers Struthers 120 years ago: “It is true this must mean an increased expenditure, because only the best trained men and women can do this work properly. But the child’s health is the most important resource in the earning capacity of the man.”

Further studies in this field of research are recommended in order to generalize and reinforce our results

## Figures and Tables

**Table 1 healthcare-10-01228-t001:** Sample characteristics.

	*n* (%)
School	
Public	330 (93.2)
Private	24 (6.8)
School catchment area population density	
up to1999 residents	22 (6.3)
2000–9999 residents	56 (15.9)
10,000–250,000 residents	178 (50.6)
more than 250,000 residents	96 (27.3)
Number of children in class, mean (SD)	20.7 (5.5)
Child’s age, mean (SD)	11.0 (4.0)
Child’s age at diagnosis, mean (SD)	5.9 (3.4)
Child’s gender	
Male	156 (44.3)
Female	196 (55.7)
Father’s age, mean (SD)	45.5 (6.8)
Mother’s age, mean (SD)	42.5 (5.3)
Father’s working status	
Full time	304 (88.4)
Part time	12 (3.5)
Unemployed	18 (5.2)
Retired	10 (2.9)
Father’s educational level	
Primary school	6 (1.7)
Middle school	46 (13.2)
High school	94 (27)
College	70 (20.1)
University	90 (25.9)
Postgraduate studies	42 (12.1)
Mother’s working status	
Full time	172 (48.6)
Part time	58 (16.4)
Unemployed	120 (33.9)
Retired	4 (1.1)
Mother’s educational level	
Primary school	4 (1.1)
Middle school	12 (3.4)
High school	72 (20.3)
College	92 (26)
University	122 (34.5)
Postgraduate studies	52 (14.7)
Parental family status	
Living together	306 (86.9)
Living separately by choice (separated/divorced)	38 (10.8)
Living separately from need (e.g., parent working in another city)	4 (1.1)
Widowed	4 (1.1)
Siblings	264 (74.6)

**Table 2 healthcare-10-01228-t002:** Diabetic management in school.

	*n* (%)
Child attending extended school *	38 (11)
If yes, who manages his/her diabetes?	
Parent	8 (22.2)
Teacher	2 (5.6)
Principal	4 (11.1)
Child	12 (33.3)
School nurse	2 (5.6)
Other	8 (22.4)
If not, reason for not attending extended school has to do with the diabetes	90 (34.4)
Where is your child allowed to measure glucose levels?	
Anywhere in the school	270 (77.1)
In his/her classroom	52 (14.9)
In the nurse’s office	42 (12)
Other	32 (9.1)
Where is your child allowed to inject insulin?	
Anywhere in the school	240 (68.6)
In his/her classroom	52 (14.9)
In the nurse’s office	50 (14.3)
Other	38 (10.9)
Is your child allowed to use the restroom when needed?	336 (94.4)
Is your child allowed to have a snack during class?	332 (94.9)
Do you provide all medical supplies and snacks for your child’s diabetes at school?	
No	16 (4.5)
Yes	304 (85.9)
Usually	34 (9.6)
School nurse in school	208 (58.8)
School nurse office in school	66 (21.9)
The school nurse is in school	
Full-time	164 (79.6)
Part-time	42 (20.4)
Knowledge of designated diabetes caregiver during school hours	248 (89.9)
How important is the presence of a school nurse?	
Not at all	2 (0.6)
A little	4 (1.2)
Moderately	18 (5.3)
A lot	50 (14.8)
Very	264 (78.1)

* Extended school: extracurricular activities and homework after normal school hours.

**Table 3 healthcare-10-01228-t003:** Participants’ satisfaction with the management of low or high glucose incidents during school hours, the child’s school performance in the previous year, the most recent levels of HbA1c and correlation of children’s absences due to diabetes with the existence of a school nurse’s office.

	School Nurse Office in School		*p* ‡‡
	No	Yes	
	Median (IQR)	Median (IQR)	
Level of satisfaction with management of incidents during school hours of your child having blood glucose of *less than* 70 mg/dL with symptoms?	3 (2–4)	3 (3–4)	0.011
Level of satisfaction with management of incidents during school hours of your child having glucose of *more than* 250 mg/dL with symptoms?	3 (1–4)	3 (2–4)	0.019
How was the child’s school performance in the previous year characterized?	4 (3–5)	5 (4–5)	<0.001
Most recent levels of HbA1c	7 (6.5–7.6)	7 (6.2–7.2)	0.043
School absences during previous year due to diabetes	7 (3–20)	5 (2–7)	0.023

IQR: interquartile range; ‡‡ Mann–Whitney test.

**Table 4 healthcare-10-01228-t004:** Association of participants’ feeling safe for their child during school hours with information regarding management of their child’s diabetes.

	How Safe Do You Think Your Child Is during School Hours?		
	Not at All/ A Little/Moderately	Very/Very Much	Extremely
	*n* (%)	*n* (%)	*p*
Is the school nurse providing most of the care for your child’s diabetes during a normal school day?			
No	166 (74.1)	58 (25.9)	**<0.001 +**
Yes	30 (23.1)	100 (76.9)	
How do you evaluate the school staff’s ability to manage your child’s diabetes? (median (IQR))			
Teacher	0 (0–1)	1 (0–2.5)	**0.009 ‡‡**
School nurse	2 (1–4)	4 (4–4)	**<0.001 ‡‡**
Coach or trainer	0 (0–1)	0 (0–1)	0.985 ‡‡
Principal	0 (0–1)	0 (0–1)	0.878 ‡‡
Other	3 (2–4)	3 (3–4)	0.770 ‡‡
Does your child know how to measure his/her blood glucose without any supervision or help?			
No	24 (44.4)	30 (55.6)	0.087 +
Yes	170 (57)	128 (43)	
Is your child’s glucose measured at school?			
No	16 (72.7)	6 (27.3)	0.098 +
Yes	178 (54.6)	148 (45.4)	
Does your child know how to inject insulin without any supervision or help?			
No	40 (40)	60 (60)	**<0.001 +**
Yes	156 (61.4)	98 (38.6)	
Is the school nurse trained to take care of your child’s diabetes?			
No	44 (57.9)	32 (42.1)	
Yes	150 (55.1)	122 (44.9)	0.670 +
Would you allow the school nurse to take care of or help your child with their diabetes, assuming he/she was adequately trained?			
No	40 (66.7)	20 (33.3)	
Yes	154 (53.5)	134 (46.5)	0.061 +

+ Pearson’s chi-square test; ‡‡ Mann–Whitney test.

**Table 5 healthcare-10-01228-t005:** Multivariate logistic regression results, with feeling very/extremely safe and satisfied with the care provided to their child as dependent variables.

Dependent Variable		OR (95% CI) +	*p*
Feeling very/extremely safe with the care provided to their child during a normal school day			
Child’s age		0.82 (0.73–0.92)	**<0.001**
School nurse’s ability to take care of your child’s diabetes during a normal school day		6.36 (3.77–10.73)	**<0.001**
School nurse office in school	No (reference)		
	Yes	3.64 (1.53–8.67)	**0.004**
Being very/extremely satisfied with the care provided to your child during a normal school day			
Child’s age		0.77 (0.68–0.86)	**<0.001**
School nurse’s ability to take care of your child’s diabetes during a normal school day		5.7 (3.39–9.59)	**<0.001**
School nurse office in school	No (reference)		
	Yes	2.4 (1.03–5.6)	**0.044**
Father’s educational level	Primary/Middle school (reference)		
	High school/College	3.71 (1.01–13.63)	**0.048**
	University/Postgraduate studies	2.08 (0.57–7.63)	0.271

+ Odds Ratio (95% Confidence Interval).

## Data Availability

The data presented in this study are available on request from the corresponding author. The data are not publicly available due to privacy reasons.

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
