# Peer review of "Care and Safety of Schoolchildren with Type 1 Diabetes Mellitus: Parental Perceptions of the School Nurse Role"

_healthcare, 2022, doi:10.3390/healthcare10071228_

Round 1

Reviewer 1 Report

Dear authors, thank you for submitting your paper to this journal. This is an interesting paper and I hope that the comments below will help you improve it.

In your Abstract, I would suggest to replace higher with advanced in the following sentence: "...higher school nurses’ diabetic care skills".

In the last sentence of the abstract replace students with schoolchildren.

In your Introduction,

Lines 61-63 could probably be moved at the end of this paragraph.

Line 79, what do you mean by special education? is it special education institutes?

Line 84, I would suggest to start the sentence: There are limited studies exploring ..... in Greece.

Line 87, better to say the diabetic child in school (generally avoid the term student as student ages vary in  different countries).

Under 2.1 Instrument, can you provide any information about validity and reliability of the questionnaires in their original language? What are the scales used to respond to these questionnaires (Likert type scales)? Can you also provide additional information on the "Cultural weighting of both the questionnaires for adaptation to the Greek language"? What was the process you followed? Furthermore, what type of modifications were made to the questionnaires?

On line 115, can you provide the age range of children attending primary/secondary school, to understand better the context?

For Table 2, can you add some explanation on what "extended school" is?

The sentence starting at line 181 to 185 is not very clear. The same goes for the sentence on lines 189-193 (is it the parent's assessment or parent's perception?)

In line 202, is it the more able or the more qualified?

The first sentence opening the Discussion needs some editing.

I am not sure what you mean here: In Greece, the ratio of school nurses / students with diabetes is basically 1:1... Is this correct?

Review the following sentences, lines 202-222: "Wilt on the other hand, linked the security and satisfaction.... higher nursing / student ratios [9]." 

Line 224, it is better to say: The literature confirms...

Line 226, is it about teachers' safety or the children's safety?

Avoid long sentences, for example the one in lines 264-268 (challenging for the reader).

Is this sentence correct: "Daily number of absences due to diabetes in this study was 6 days." is this per month, per year?

Probably it would be best to reconsider this in line 290:  "the institution of the school nurse does not exist" Is it an institution? they do not have a permanent school nurse post?

Consider the limitations of the study widely, not just from the point of recruitment/participants. For example, limitations of cross sectional studies, use of questionnaires that do not allow for explanations etc.

Under conclusions, I felt that some recommendations are unrealistic and would required further investigation before they are suggested. Each school having a school nurse would be ideal but what about the costs involved? Other strategies may be more cost-effective.

Author Response

Dear reviewer 1,

Thank you very much for your comments.

We have attached our response to them

yours sincerly

Reviewer 2 Report

Thank you for allowing me to review this manuscript. This manuscript entitled "Care and safety of schoolchildren with type 1 diabetes mellitus: parents' perceptions of the role of the school nurse", aims to investigate the perceptions of parents about the care and safety of their children with DM1 in the school.

It is an interesting and highly relevant article today, although it has several limitations that make it suitable for publication in this journal. These limitations are detailed below:

- The citation regulations required by the journal are not strictly followed, since there are errors in the citations. It is necessary to review the citations and adapt them to the regulations indicated by the journal. For example, in the lines: 42, 44, 209, 225.

- I would recommend that the authors justify the novelty and relevance of the study being carried out in more detail in the Introduction

- It would be interesting to include in more detail information on the local context for data collection. Why is it advisable to collect data there? What are the reasons for doing it?

- In the material and methods section, it would be interesting to provide more information on the questionnaires used for data collection. I would recommend the authors to specify whether the questionnaires are validated and the psychometric properties of said questionnaires. Also, it would be important to point out the inclusion and application requirements among the participants. Another aspect that would be important to point out in the material and methods section is the study design.

- In the results section, I would recommend putting a table footer in the tables, where the acronyms of the tables will be specified.

- The conclusions are very elaborated, they are clear and precise. It also reflects the importance and impact of the subject of study in the clinical setting. However, I would recommend the authors to include a proposal for the future to continue in this field of research.

Author Response

Dear reviewer 2,

Thank you for your kind comments.

Please find attached our response.

yours sincerely
